# Reliable Adversarial Distillation with Unreliable Teachers

**Jianing Zhu**[1]    **Jiangchao Yao**[2]    **Bo Han**[1,†]    **Jingfeng Zhang**[3]
**Tongliang Liu**[4]    **Gang Niu**[3]    **Jingren Zhou**[2]    **Jianliang Xu**[1]    **Hongxia Yang**[2]

[1]Hong Kong Baptist University [2]Alibaba Group
[3]RIKEN Center for Advanced Intelligence Project [4]The University of Sydney

{csjnzhu, bhanml, xujl}@comp.hkbu.edu.hk
{jiangchao.yjc, jingren.zhou, yang.yhx}@alibaba-inc.com
jingfeng.zhang@riken.jp  tongliang.liu@sydney.edu.au  gang.niu.ml@gmail.com

## Abstract

In ordinary distillation, student networks are trained with *soft labels* (SLs) given by *pretrained* teacher networks, and students are expected to improve upon teachers since SLs are *stronger* supervision than the original *hard labels*. However, when considering *adversarial robustness*, teachers may become unreliable and adversarial distillation may not work: teachers are pretrained on their own adversarial data, and it is too demanding to require that teachers are also good at every adversarial data queried by students. Therefore, in this paper, we propose reliable *introspective adversarial distillation* (IAD) where students *partially* instead of *fully* trust their teachers. Specifically, IAD distinguishes between three cases given a query of a *natural data* (ND) and the corresponding *adversarial data* (AD): (a) if a teacher is good at AD, its SL is fully trusted; (b) if a teacher is good at ND but not AD, its SL is partially trusted and the student also takes its own SL into account; (c) otherwise, the student only relies on its own SL. Experiments demonstrate the effectiveness of IAD for improving upon teachers in terms of adversarial robustness.

## 1 Introduction

Deep Neural Networks (DNNs) have shown excellent performance on a range of tasks in computer vision (He et al., 2016) and natural language processing (Devlin et al., 2019). Nevertheless, Szegedy et al. (2014); Goodfellow et al. (2015) demonstrated that DNNs could be easily fooled by adding a small number of perturbations on natural examples, which increases the concerns on the robustness of DNNs in the trustworthy-sensitive areas, *e.g.,* finance (Kumar et al., 2020) and autonomous driving (Litman, 2017). To overcome this problem, adversarial training (Goodfellow et al., 2015; Madry et al., 2018) is proposed and has shown effectiveness to acquire the adversarially robust DNNs.

Most existing adversarial training approaches focus on learning from data directly. For example, the popular adversarial training (AT) (Madry et al., 2018) leverages multi-step projected gradient descent (PGD) to generate the adversarial examples and feed them into the standard training. Zhang et al. (2019) developed TRADES on the basis of AT to balance the standard accuracy and robust performance. Recently, there are several methods under this paradigm are developed to improve the model robustness (Wang et al., 2019; Alayrac et al., 2019; Carmon et al., 2019; Zhang et al., 2020; Jiang et al., 2020; Ding et al., 2020; Du et al., 2021; Zhang et al., 2021). However, directly learning from the adversarial examples is a challenging task on the complex datasets since the loss with hard labels is difficult to be optimized (Liu et al., 2020), which limits us to achieve higher robust accuracy.

To mitigate this issue, one emerging direction is distilling robustness from the adversarially pre-trained model intermediately, which has shown promise in the recent study (Zi et al., 2021; Shu et al., 2021). For example, Ilyas et al. (2019) used an adversarially pre-trained model to build a "robustified" dataset to learn a robust DNN. Fan et al. (2021); Salman et al. (2020) explored to boost the model

---

[†]Corresponding author (bhanml@comp.hkbu.edu.hk).

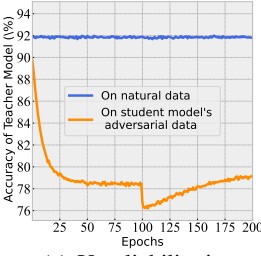 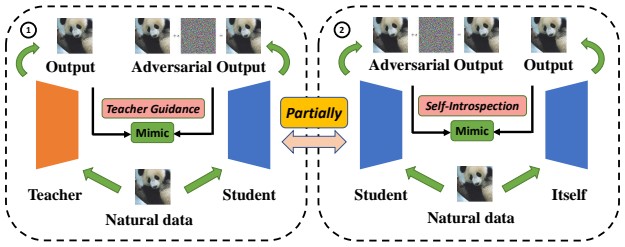

(a) Unreliability issue

(b) Overview of Introspective Adversarial Distillation (IAD)

Figure 1: **(a) Unreliability issue**: Comparison of the teacher model's accuracy on the natural data and the student model's adversarial data from *CIFAR-10* datasets. Different from the ordinary distillation in which the natural data is unchanged and the teacher model has the consistent standard performance, the teacher model's robust accuracy on the student model's adversarial data is decreasing during distillation, which means the guidance of the teacher model in adversarial distillation is progressively unreliable. **(b) Overview of Introspective Adversarial Distillation (IAD)**: The student *partially trusts the teacher guidance* and *partially trusts self-introspection* during adversarial distillation. Specifically, the student model generates the adversarial data by itself and mimics the outputs of the teacher model and itself partially in IAD. Note that, the concrete intuition about the student introspection is referred to the analysis and the evidence in Figure 2 and Figure 3.

robustness through fine-tuning or transfer learning from adversarially pre-trained models. Goldblum et al. (2020)and Chen et al. (2021) investigated distilling the robustness from adversarially pre-trained models, termed as *adversarial distillation* for simplicity, where they encouraged student models to mimic the outputs (i.e., soft labels) of the adversarially pre-trained teachers.

However, one critical difference is: in the conventional distillation, the teacher model and the student model share the natural training data; while in the adversarial distillation, the adversarial training data of the student model and that of the teacher model are egocentric (respectively generated by themselves) and becoming more adversarial challenging during training. Given this distinction, are the soft labels acquired from the teacher model in adversarial distillation always reliable and informative guidance? To answer this question, we take a closer look at the process of adversarial distillation. As shown in Figure 1(a), we discover that along with the training, the teacher model progressively fails to give a correct prediction for the adversarial data queried by the student model. The reason could be that with the students being more adversarially robust and thus the adversarial data being harder, it is too demanding to require the teachers become always good at every adversarial data queried by the student model, as the teacher model has never seen these data in its pre-training. In contrast, for the conventional distillation, student models are expected to distill the *"static"* knowledge from the teacher model, since the soft labels for the natural data from the teacher model are always fixed.

The observation in Figure 1(a) raises the challenge: *how to conduct reliable adversarial distillation with unreliable teachers*? To solve this problem, we can categorize the training data according to the prediction on natural and student generated adversarial data into three cases. First, if the teacher model can correctly classify both natural and adversarial data, it is reliable; Second, if the teacher model can correctly classify the natural but not adversarial data, it should be partially trusted, and the student model is suggested to trust itself to enhance model robustness as the adversarial regularization (Zhang et al., 2019); Third, if the teacher model cannot correctly classify both natural and adversarial data, the student model is recommended to trust itself totally. According to this intuition, we propose an Introspective Adversarial Distillation (IAD) to effectively utilize the knowledge from an adversarially pre-trained teacher model. The framework of our proposed IAD can be seen in Figure 1(b). Briefly, the student model is encouraged to *partially* instead of *fully* trust the teacher model, and gradually trust itself more as being more adversarially robust. We conduct extensive experiments on the benchmark *CIFAR-10/CIFAR-100* and the more challenging *Tiny-ImageNet* datasets to evaluate the efficiency of our IAD. The main contributions of our work can be summarized as follows.

1. We take a closer look at adversarial distillation under the teacher-student paradigm. Considering adversarial robustness, we discover that the guidance from the teacher model is progressively unreliable along with the adversarial training.

2. We construct the reliable guidance for adversarial distillation by flexibly utilizing the robust knowledge from the teacher model: (a) if a teacher is good at adversarial data, its soft labels can be fully trusted; (b) if a teacher is good at natural data but not adversarial data, its soft

labels should be partially trusted and the student also takes its own soft labels into account; (c) otherwise, the student only relies on its own soft labels.

3. We propose an Introspective Adversarial Distillation (IAD) to automatically realize the intuition of the previous reliable guidance during the adversarial distillation. The experimental results confirmed that our approach can improve adversarial robustness across a variety of training settings and evaluations, and also on the challenging (consider adversarial robustness) datasets (e.g., *CIFAR-100* (Krizhevsky, 2009) and *Tiny-ImageNet* (Le & Yang, 2015)) or using large models (e.g., *WideResNet* (Zagoruyko & Komodakis, 2016)).

## 2 RELATED WORK

### 2.1 ADVERSARIAL TRAINING.

Adversarial examples (Goodfellow et al., 2015) motivate many defensive approaches developed in the last few years. Among them, adversarial training has been demonstrated as the most effective method to improve the robustness of DNNs (Cai et al., 2018; Wang et al., 2020; Jiang et al., 2020; Chen et al., 2021; Sriramanan et al., 2021). The formulation of the popular AT (Madry et al., 2018) and its variants can be summarized as the minimization of the following loss:

$$\min_{f_\theta \in \mathcal{F}} \frac{1}{n} \sum_{i=1}^{n} \ell(f_\theta(\tilde{x}_i), y_i), \tag{1}$$

where $n$ is the number of training examples, $\tilde{x}_i$ is the adversarial example within the $\epsilon$-ball (bounded by an $L_p$-norm) centered at natural example $x_i$, $y_i$ is the associated label, $f_\theta$ is the DNN with parameter $\theta$ and $\ell(\cdot)$ is the standard classification loss, e.g., the cross-entropy loss. Adversarial training leverages adversarial examples to smooth the small neighborhood, making the model prediction locally invariant. To generate the adversarial examples, AT employs a PGD method (Madry et al., 2018). Concretely, given a sample $x^{(0)} \in \mathcal{X}$ and the step size $\beta > 0$, PGD recursively searches

$$\tilde{x}^{(t+1)} = \Pi_{\mathcal{B}[\tilde{x}^{(0)}]} \big( \tilde{x}^{(t)} + \beta \operatorname{sign}(\nabla_{\tilde{x}^{(t)}} \ell(f_\theta(\tilde{x}^{(t)}), y)) \big), \tag{2}$$

until a certain stopping criterion is satisfied. In Eq. equation 2, $t \in \mathbb{N}$, $\ell$ is the loss function, $\tilde{x}^{(t)}$ is adversarial data at step $t$, $y$ is the corresponding label for natural data, and $\Pi_{\mathcal{B}_\epsilon[\tilde{x}_0]}(\cdot)$ is the projection function that projects the adversarial data back into the $\epsilon$-ball centered at $\tilde{x}^{(0)}$.

### 2.2 KNOWLEDGE DISTILLATION

The idea of distillation from other models can be dated back to (Craven & Shavlik, 1996), and re-introduced by (Hinton et al., 2015) as knowledge distillation (KD). It has been widely studied in recent years (Yao et al., 2021) and works well in numerous applications like model compression and transfer learning. For adversarial defense, a few studies have explored obtaining adversarial robust models by distillation. Papernot et al. (2016) proposed defensive distillation which utilizes the soft labels produced by a standard pre-trained teacher model, while this method is proved to be not resistant to the C&W attacks (Carlini & Wagner, 2017); Goldblum et al. (2020) combined AT with KD to transfer robustness to student models, and they found that the distilled models can outperform adversarially pre-trained teacher models of identical architecture in terms of adversarial robustness; Chen et al. (2021) utilized distillation as a regularization for adversarial training, which employed robust and standard pre-trained teacher models to address the robust overfitting (Rice et al., 2020).

Nonetheless, all these related methods fully trust teacher models and do not consider that whether the guidance of the teacher model in distillation is reliable or not. In this paper, different from the previous studies, we find that the teacher model in adversarial distillation is not always trustworthy. Formally, adversarial distillation suggests to minimize $\mathbb{E}_{\tilde{x} \in \mathcal{B}[x]} \left[ \ell_{kl}(S(\tilde{x}|\tau) || T(\tilde{x}|\tau)) \right]$, where $T(\tilde{x}|\tau)$ is not a constant soft supervision along with the adversarial training and affected by the adversarial data generated by the dynamically evolved student network. Based on that, we propose reliable IAD to encourage student models to partially instead of fully trust teacher models, which effectively utilizes the knowledge from the adversarially pre-trained models.

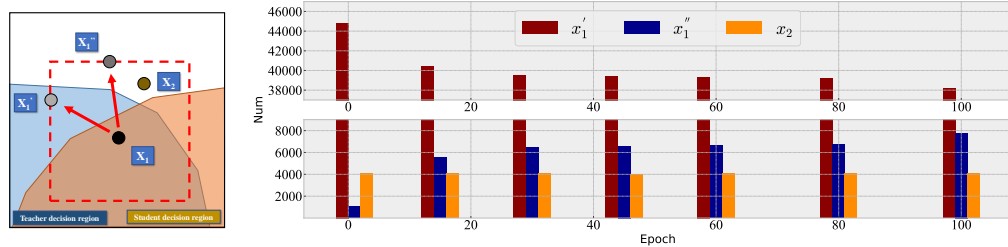

(a) Toy illustration        (b) Number changes on the three kinds of data during distillation

Figure 2: **(a) Toy illustration**: An illustration of three situations about the prediction of the teacher model. The blue/orange areas represent the decision region (in which the model can give the correct predictions) of the teacher/student model, and the red dashed box represents the unit-norm ball of AT. 1) $x_1'$. The generated adversarial example from the natural example $x_1$ is located in the blue area, where the teacher model can correctly predict; 2) $x_1''$. The generated adversarial example from $x_1$ is out of both orange and blue areas, where the teacher model has the wrong prediction; 3) $x_2$. The natural example that the teacher model cannot correctly predict. **(b) Number changes on the three kinds of data**: We trace the number of three types of data during adversarial distillation on *CIFAR-10* dataset. Noted that, different from the consistent number of examples like $x_2$, the number of examples like $x_1'$ is *decreasing* and that of examples like $x_1''$ is *increasing* during adversarial distillation.

## 3 A CLOSER LOOK AT ADVERSARIAL DISTILLATION

In Section 3.1, we discuss the unreliable issue of adversarial distillation, i.e., the guidance of the teacher model is progressively unreliable along with adversarial training. In Section 3.2, we partition the training examples into three parts and analyze them part by part. Specifically, we expect that the student model should *partially* instead of *fully* trust the teacher model and gradually trust itself more along with adversarial training.

### 3.1 FULLY TRUST: PROGRESSIVELY UNRELIABLE GUIDANCE

As aforementioned in the Introduction, previous methods (Goldblum et al., 2020; Chen et al., 2021) fully trust the teacher model when distilling robustness from adversarially pre-trained models. Taking Adversarial Robust Distillation (ARD) (Goldblum et al., 2020) as an example, we illustrate its procedure in the left part of Figure 1(b): the student model generates its adversarial data and then optimizes the prediction of them to mimic the output of the teacher model. However, although the teacher model is well optimized on the adversarial data queried by itself, we argue that it might not always be good at the more and more challenging adversarial data queried by the student model.

As shown in Figure 1(a), different from the ordinary distillation in which the teacher model has the consistent standard performance on the natural data, its robust accuracy on the student model's adversarial data is decreasing during distillation. The guidance of the teacher model gradually fails to give the correct output on the adversarial data queried by the student model.

### 3.2 PARTIALLY TRUST: CONSTRUCTION OF RELIABLE GUIDANCE

The unreliable issue of the teacher model in adversarial distillation raises the challenge of how to conduct *reliable adversarial distillation with unreliable teachers*? Intuitively, this requires us to re-consider the guidance of adversarially pre-trained models along with the adversarial training. For simplicity, we use $\mathcal{T}(x)$ ($\mathcal{T}(\tilde{x})$) to represent the predicted label of the teacher model on the natural (adversarial) examples, and use $y$ to represent the targeted label. We partition the adversarial samples into following parts as shown in the toy illustration (Figure 2(a)), and analyze them part by part.

1) $\mathcal{T}(x) = y \cap \mathcal{T}(\tilde{x}) = y$: It can be seen in Figure 2(a) that this part of data whose adversarial variants like $x_1'$ is the most trustworthy among the three parts, since the teacher model performs well on both natural and adversarial data. In this case, we could choose to trust the guidance of the teacher model on this part of the data. However, as shown in Figure 2(b), we find that the sample number of this part is decreasing along with the adversarial training. That is, what we can rely on from the teacher model in adversarial distillation is progressively reduced.

2) $\mathcal{T}(x) = y \cap \mathcal{T}(\tilde{x}) \neq y$: In Figure 2(b), we also check the number change of the part of data whose adversarial variants like $x_1''$. Corresponding to the previous category, the number of this kind of data is increasing during distillation. Since the teacher model's outputs on the small neighborhood of the queried natural data are not always correct, its knowledge may not be robust and the guidance for the student model is not reliable. Think back to the reason for the decrease in the robust accuracy of the teacher model, the student model itself may also be trustworthy since it becomes gradually adversarial robust during distillation.

3) $\mathcal{T}(x) \neq y \cap \mathcal{T}(\tilde{x}) \neq y$: As for the data which are like $x_2$ in Figure 2(a), the guidance of the teacher model is totally unreliable since the predicted labels on the natural data are wrong. The student model may also trust itself to encourage the outputs to mimic that of their natural data rather than the wrong outputs from the teacher model. First, it removes the potential threat that the teacher's guidance may be a kind of noisy labels for training. Second, as an adversarial regularization (Zhang et al., 2019), it can improve the model robustness through enhancing the stability of the model's outputs on the natural and the corresponding adversarial data.

4) $\mathcal{T}(x) \neq y \cap \mathcal{T}(\tilde{x}) = y$: Considering the generation process of the adversarial data, *i.e.,* $\tilde{x}^* = \arg\max_{\tilde{x} \in \mathcal{B}_\epsilon(x)} \ell(f(\tilde{x}), y)$, Once the original prediction is wrong, *i.e.,*, $T(x) \neq y$, the generation of $\tilde{x}^*$ only make the prediction worse. Thus, this group doesn't exist.

To sum up, we suggest employing reliable guidance from the teacher model and encouraging the student model to trust itself more as the teacher model's guidance being progressively unreliable and the student model gradually becoming more adversarially robust.

## 4 INTROSPECTIVE ADVERSARIAL DISTILLATION

Based on previous analysis about the adversarial distillation, we propose the Introspective Adversarial Distillation (IAD) to better utilize the guidance from the adversarially pre-trained model. Concretely, we have the following KD-style loss, but composite with teacher guidance and student introspection.

$$\ell_{IAD} = \underbrace{\mathcal{O}(AD_i; \alpha)}_{\textbf{Label} \& \textbf{ Teacher Guidance}} + \gamma \underbrace{\ell_{KL}(S(\tilde{x}|\tau)||S(x|\tau)))}_{\textbf{Student Introspection}}, \tag{3}$$

where $\mathcal{O}(AD_i; \alpha)$ is the previous adversarial distillation baseline, e.g., ARD (Goldblum et al., 2020) or AKD$^2$ (Chen et al., 2021), weighted by the hyper-parameter $\alpha$[1], $\gamma$ is a weight for the student introspection, $S(\cdot|\tau)$ is a Softmax operator with the temperature $\tau \in (0, +\infty)$, *e.g.,* $S(x_k|\tau) = \frac{\exp(x_k/\tau)}{\sum_{k'} \exp(x_{k'}/\tau)}$, $S(\cdot)$ is the conventional Softmax with the temperature $\tau = 1$, $T(\cdot|\tau)$ is the tempered variant of the teacher output $T(\cdot)$, $\tilde{x}$ is the adversarial data generated from the natural data $x$, $y$ is the hard label and $\ell_{CE}$ is the Cross Entropy loss and $\ell_{KL}$ is the KL-divergence loss. As for the annealing parameter $\alpha \in [0, 1]$ that is used to balance the effect of the teacher model in adversarial distillation, based on the analysis about the reliability of adversarial supervision in Section 3, we define it as,

$$\alpha = (P_T(\tilde{x}|y))^\beta, \tag{4}$$

where $P_T(\cdot|y)$ is the prediction probability of the teacher model about the targeted label $y$ and $\beta$ is a hyperparameter to sharpen the prediction. The intuition behind IAD is to calibrate the guidance from the teacher model automatically based on the prediction of adversarial data. Our $\alpha$ naturally corresponds to the construction in Section 3.2, since the prediction probability of the teacher model for the adversarial data can well represent the categorical information. As for $\beta$, we have plot the specific values of $\alpha$ under its adjustment in the left of Figure 4.

Intuitively, the student model can trust the teacher model when $\alpha$ approaches 1, which means that the teacher model is good at both natural and adversarial data. However, when $\alpha$ approaches 0, it corresponds that the teacher model is good at natural but not adversarial data, or even not good at both, and thus the student model should take its self-introspection into account. In Figure 3, we check the reliability of the student model itself. According to the left panel of Figure 3, we can see that the student model is progressively robust to the adversarial data. And if we incorporate the student introspection into the adversarial distillation, the results in the middle of Figure 3 confirms its

---

[1] Note that, we do not use $\alpha$ when $AD_i$ is ARD. Please refer to the Appendix A.2 for the ablation study.

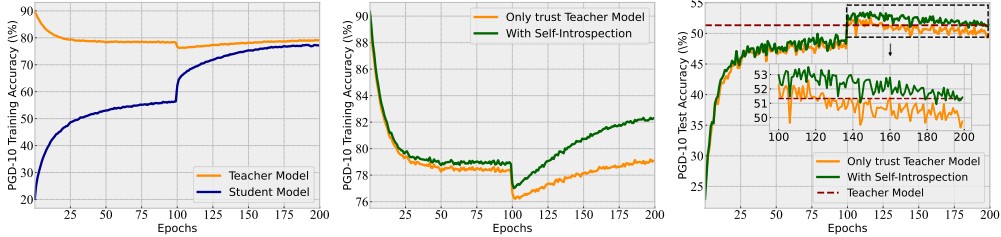

Figure 3: Reliability certification about self-introspection of the student model: *Left*, PGD-10 training accuracy of teacher/student model; *Middle*, PGD-10 training accuracy of teacher model and that combined with the self-introspection of the student model; *Right*, PGD-10 test accuracy of the student model which only trusts the soft labels and the student model which also considers self-introspection.

---

**Algorithm 1** Introspective Adversarial Distillation (IAD)

---

**Input:** student model $S$, teacher model $T$, training dataset $D = \{(x_i, y_i)\}_{i=1}^n$, learning rate $\eta$, number of epochs $N$, batch size $m$, number of batches $M$, temperature parameter $\tau$, the annealing parameter on teacher model's predicted probability $\alpha$, adjustable parameter $\lambda$, $\lambda_1$, $\lambda_2$, $\lambda_3$ and $\gamma$.
**Output:** adversarially robust model $S_r$
**for** epoch $= 1, \ldots, N$ **do**
    **for** mini-batch $= 1, \ldots, M$ **do**
        Sample a mini-batch $\{(x_i, y_i)\}_{i=1}^m$ from $D$
        **for** $i = 1, \ldots, m$ (in parallel) **do**
            Obtain adversarial data $\tilde{x}_i$ of $x_i$ by PGD based on Eq. equation 2.
            Compute $\alpha$ for each adversarial data based on Eq. equation 4.
        **end for**
        IAD-I: $\theta \leftarrow \theta - \eta \nabla_\theta \left\{ \begin{array}{l} \lambda \ell_{CE}(S(x), y) + (1 - \lambda) \cdot \tau^2 \cdot \\ (\ell_{KL}(S(\tilde{x}|\tau)||T(x|\tau)) + \gamma \ell_{KL}(S(\tilde{x}|\tau)||S(x|\tau))) \end{array} \right\}$
        **or**
        IAD-II: $\theta \leftarrow \theta - \eta \nabla_\theta \left\{ \begin{array}{l} \lambda_1 \ell_{CE}(S(\tilde{x}), y) + \lambda_2 \cdot \tau^2 \cdot \\ (\alpha \cdot \ell_{KL}(S(\tilde{x}|\tau)||T_{at}(\tilde{x}|\tau)) + \lambda_3 \tau^2 \ell_{KL}(S(\tilde{x}|\tau)||T_{st}(\tilde{x}|\tau)) + \\ \gamma \ell_{KL}(S(\tilde{x}|\tau)||S(x|\tau))) \end{array} \right\}$
    **end for**
**end for**

---

potential benefits to improve the accuracy of the guidance. Moreover, as shown in the right panel of Figure 3, adding self-introspection results in better improvement in model robustness compared to only using the guidance of the teacher model. Therefore, $\ell_{IAD}$ automatically encourages the outputs of the student model to mimic more reliable guidance in adversarial distillation.

Algorithm 1 summarizes the implementation of Introspective Adversarial Distillation (IAD). Specifically, IAD first leverages PGD to generate the adversarial data for the student model. Secondly, IAD computes the outputs of the teacher model and the student model on the natural data. Then, IAD mimics the outputs of the student model with that of itself and the teacher model partially based on the probability of the teacher model on the adversarial data.

**Warming-up period.** During training, we add a warming-up period to activate the student model, where $\alpha$ (in Eq. equation 3) is hardcoded to 1. This is because the student itself is not trustworthy in the early stage (refer to the left panel of Figure 3). Through that, we expect the student model to first evolve into a relatively reliable learner and then conducts the procedure of IAD.

### 4.1 COMPARISON WITH RELATED METHODS

In this section, we discuss the difference between IAD and other related approaches in the perspective of the loss functions. Table 1 summarizes all of them.

As shown in Table 1, AT (Madry et al., 2018) utilizes the hard labels to supervise adversarial training; TRADES (Zhang et al., 2019) decomposes the loss function of AT into two terms, one for standard training and the other one for adversarial training with the soft supervision; Motivated by KD (Hinton et al., 2015), Goldblum et al. (2020) proposed ARD to conduct the adversarial distillation, which fully

Table 1: Loss comparison. The checkmark means that the term is considered in the model objective.

| Knowledge | Label | | Self | Teacher | | |
|---|---|---|---|---|---|---|
| Loss Term | $\ell_{CE}(S(x),y)$ | $\ell_{CE}(S(\tilde{x}),y)$ | $\ell_{KL}(S(\tilde{x}|\tau)||S(x|\tau))$ | $\ell_{KL}(S(\tilde{x}|\tau)||T_{at}(x|\tau))$ | $\ell_{KL}(S(\tilde{x}|\tau)||T_{at}(\tilde{x}|\tau))$ | $\ell_{KL}(S(\tilde{x}|\tau)||T_{st}(\tilde{x}|\tau))$ |
| AT | | ✓ | | | | |
| TRADES | ✓ | | ✓ | | | |
| ARD | ✓ | | | ✓ | | |
| IAD-I | ✓ | | ✓ | ✓ | | |
| AKD$^2$ | | ✓ | | | ✓ | ✓ |
| IAD-II | | ✓ | ✓ | | ✓ | ✓ |

Table 2: Test accuracy (%) on *CIFAR-10/CIFAR-100* datasets using *ResNet-18*.

| Datasets | | CIFAR-10 | | | | | CIFAR-100 | | | | |
|---|---|---|---|---|---|---|---|---|---|---|---|
| Teacher | Method | Natural | FGSM | PGD-20 | CW$_\infty$ | AA | Natural | FGSM | PGD-20 | CW$_\infty$ | AA |
| AT | AT | 83.06% | 63.53% | 50.21% | 49.22% | 46.70% | 56.21% | 33.94% | 24.60% | 23.35% | 21.55% |
| | ARD | 83.13% | **64.11%** | 51.36% | 50.37% | 48.05% | 56.65% | 35.82% | 27.04% | 25.30% | 23.34% |
| | AKD$^2$ | **83.52%** | 63.91% | 51.36% | 50.36% | 48.08% | 56.46% | 35.76% | 27.18% | 25.33% | 23.47% |
| | **IAD-I** | 82.09% | 63.20% | **52.14%** | **50.74%** | **48.66%** | 55.53% | 35.69% | 27.40% | **25.80%** | 23.66% |
| | **IAD-II** | 83.21% | 63.54% | 51.85% | 50.67% | 48.58% | **57.26%** | **35.94%** | **27.50%** | 25.68% | **24.06%** |
| TRADES | TRADES | 81.26% | 63.12% | 52.98% | 50.29% | 49.47% | 54.01% | 35.23% | 28.00% | 24.26% | 23.46% |
| | ARD | 81.50% | 63.38% | 53.38% | 51.27% | 49.92% | 55.82% | **37.56%** | 30.21% | 26.76% | 25.36% |
| | AKD$^2$ | 83.49% | 64.00% | 51.89% | 50.25% | 48.52% | **57.46%** | 37.32% | 28.62% | 25.76% | 24.36% |
| | **IAD-I** | 80.63% | 63.14% | **53.84%** | **51.60%** | **50.17%** | 55.19% | 37.47% | **30.60%** | **27.24%** | **25.84%** |
| | **IAD-II** | **83.76%** | **64.17%** | 52.16% | 50.59% | 48.91% | 57.08% | 36.94% | 29.08% | 25.83% | 24.45% |

trusts the outputs of the teacher model to learn the student model. As indicated by the experiments in Goldblum et al. (2020), a larger $\lambda$ resulted in less robust student models. Thus, they generally set $\lambda = 0$ in their experiments; Chen et al. (2021) utilized distillation as a regularization to avoid the robust overfitting issue, which employed both the adversarially pre-trained teacher model and the standard pre-trained model. Thus, there are two KL-divergence loss and for simplicity, we term their method as AKD$^2$; Regarding IAD, there are two types of implementations that are respectively based on ARD or AKD$^2$. We term them as IAD-I and IAD-II, and their difference with previous ARD and AKD$^2$ is an additional self-introspection term. Besides, we also apply $\alpha$ to downweight the dependency on the term $\ell_{KL}(S(\tilde{x}|\tau)||T_{at}(\tilde{x}|\tau))$, which has been explained in previous sections.

## 5 EXPERIMENTS

We conduct extensive experiments to evaluate the effectiveness of IAD. In Section 5.1, we compare IAD with benchmark adversarial training methods (AT and TRADES) and some related methods which utilize adversarially pre-trained models via KD (ARD and AKD$^2$) on *CIFAR-10/CIFAR-100* (Krizhevsky, 2009) datasets. In Section 5.2, we compare the previous methods with IAD on a more challenging dataset *Tiny-ImageNet* (Le & Yang, 2015). In Section 5.3, the ablation studies are conducted to analyze the effects of the hyper-parameter $\beta$ and different warming-up periods for IAD.

Regarding the measures, we compute the natural accuracy on the natural test data and the robust accuracy on the adversarial test data generated by FGSM, PGD-20, and C&W attacks following (Wang et al., 2019) Moreover, we estimate the performance under AutoAttack (termed as AA).

### 5.1 EVALUATION ON *CIFAR-10/CIFAR-100* DATASETS

**Experiment Setup.** In this part, we follow the setup (learning rate, optimizer, weight decay, momentum) of Goldblum et al. (2020) to implement the adversarial distillation experiments on the *CIFAR-10/CIFAR-100* datasets. Specifically, we train ResNet-18 under different methods using SGD with 0.9 momentum for 200 epochs. The initial learning rate is 0.1 divided by 10 at Epoch 100 and Epoch 150 respectively, and the weight decay=0.0002. In the settings of adversarial training, we set the perturbation bound $\epsilon = 8/255$, the PGD step size $\sigma = 2/255$, and PGD step numbers $K = 10$. In the settings of distillation, we use $\tau = 1$ and use models pre-trained by AT and TRADES which have the best PGD-10 test accuracy as the teacher models for ARD, AKD$^2$ and our IAD. For ARD, we set its hyper-parameter $\lambda = 0$ as recommend in Goldblum et al. (2020) for gaining better robustness. For AKD$^2$, we set $\lambda_1 = 0.25$, $\lambda_2 = 0.5$ and $\lambda_3 = 0.25$ as recommended in Chen et al. (2021). For IAD-I and IAD-II, we respectively set the warming-up period as 60/80 and 40/80 epochs to train on

Table 3: Test accuracy (%) on *CIFAR-10/CIFAR-100* datasets using *WideResNet-34-10*.

| Datasets | | CIFAR-10 | | | | | CIFAR-100 | | | | |
|---|---|---|---|---|---|---|---|---|---|---|---|
| Teacher | Method | Natural | FGSM | PGD-20 | $CW_\infty$ | AA | Natural | FGSM | PGD-20 | $CW_\infty$ | AA |
| AT | AT | 85.24% | 66.07% | 53.36% | 52.85% | 50.37% | 59.75% | 39.67% | 31.14% | 29.99% | 27.46% |
| | ARD | 83.39% | 64.43% | 54.06% | 53.27% | 51.57% | 58.84% | 40.68% | 32.18% | 30.36% | 27.96% |
| | $AKD^2$ | **86.25%** | **67.65%** | 55.05% | 54.07% | 51.70% | **61.57%** | **41.14%** | 31.90% | 30.45% | 27.80% |
| | **IAD-I** | 84.21% | 66.62% | 55.07% | 54.38% | 52.11% | 57.78% | 39.95% | **32.43%** | **30.46%** | **28.39%** |
| | **IAD-II** | 85.09% | 66.54% | **55.45%** | **54.63%** | **52.29%** | 60.72% | 40.67% | 32.33% | 30.40% | 27.89% |
| TRADES | TRADES | 82.90% | 65.55% | 55.50% | 53.04% | 52.00% | 59.14% | 38.81% | 31.21% | 27.69% | 26.53% |
| | ARD | 83.60% | 65.67% | 55.32% | 53.23% | 52.08% | 58.63% | 40.20% | 32.41% | 29.86% | 27.92% |
| | $AKD^2$ | 84.95% | 67.14% | 55.38% | 53.38% | 52.12% | **61.48%** | 40.52% | 32.34% | 29.84% | 27.84% |
| | **IAD-I** | 83.06% | 66.51% | **56.17%** | **53.99%** | **52.68%** | 57.61% | 39.95% | **32.87%** | **30.13%** | **28.63%** |
| | **IAD-II** | **85.68%** | **67.39%** | 55.45% | 53.77% | 51.85% | 60.82% | 40.43% | 32.39% | 30.01% | 28.01% |

Table 4: Test accuracy (%) on *Tiny-ImageNet* dataset using *PreActive-ResNet-18*.

| Teacher | Method | Natural | FGSM | PGD-20 | $CW_\infty$ | AA |
|---|---|---|---|---|---|---|
| AT | AT | 46.16% | 28.20% | 22.16% | 19.52% | 18.04% |
| | ARD | 47.34% | 29.56% | 22.60% | 19.96% | 17.60% |
| | $AKD^2$ | 49.48% | 30.10% | 22.70% | 20.10% | 18.18% |
| | **IAD-I** | 46.14% | 29.38% | 23.40% | 20.74% | **19.16%** |
| | **IAD-II** | **49.52%** | **30.40%** | **23.42%** | **21.26%** | 18.64% |
| TRADES | TRADES | 46.12% | 27.74% | 21.00% | 16.86% | 15.86% |
| | ARD | **49.60%** | 29.98% | 22.76% | 19.16% | 17.26% |
| | $AKD^2$ | 47.72% | 28.12% | 22.10% | 18.28% | 17.13% |
| | **IAD-I** | 47.58% | 29.92% | 23.12% | 19.12% | 17.66% |
| | **IAD-II** | 48.10% | **30.54%** | **23.72%** | **19.78%** | **17.82%** |

*CIFAR-10/CIFAR-100*. Regarding the computation of $\alpha$, we set $\lambda = 0, \beta = 0.1$. For $\gamma$, we currently set $\gamma = 1 - \alpha$ and more ablation study about its setting can be found in the Appendix A.3.

**Results.** We report the results in Table 2, where the results of AT and TARDES are listed in the first and fifth rows of Table 2, and the other methods use these models as the teacher models in distillation. On *CIFAR-10* and *CIFAR-100*, we note that our IAD-I or IAD-II has obtained consistent improvements on adversarial robustness in terms of PGD-20, CW∞ and AA accuracy compared with the student models distilled by ARD or $AKD^2$ and the adversarially pre-trained teacher models. Besides, IAD-II generally performs better than IAD-I when the teacher is trained by AT, which means $AKD^2$ in this case could be a better starting point than ARD. However, when the teacher model is trained by TRAEDS, the advantage about the robustness of IAD-I over that of IAD-II is in reverse. Considering their distillation philosophy, *i.e.*, $\ell_{kl}(S(\tilde{x}|\tau)||T(x|\tau))$ of IAD-I and $\ell_{kl}(S(\tilde{x}|\tau)||T(\tilde{x}|\tau))$ of IAD-II, it might be up to which of $T(x|\tau)$ and $T(\tilde{x}|\tau)$ is more informative in adversarial distillation from the diverse teachers. The natural accuracy of IAD sometimes is lower than others, but the performance drop is not very significant compared to IAD-II.

**Experiment Setup.** In this part, we evaluate these methods by the model with a larger capacity, i.e, *WideResNet-34-10*. The teacher network is trained by AT and TRADES, following the settings of (Zhang et al., 2021). We keep the most settings of baselines same as the previous experiment. For IAD-I and IAD-II, we set $5/10$ epochs warming-up period on *CIFAR-10/CIFAR-100*.

**Results.** The results is summarized in Tables 3. Similarly, on *CIFAR-10* and *CIFAR-100*, our method can also achieve better model robustness than ARD, $AKD^2$ and the original teacher models. Moreover, our IAD methods do not sacrifice much standard performance compared with the original teacher models. Since $AKD^2$ externally utilizes a standard pre-trained teacher model, IAD-II can achieve the consistently better natural performance than IAD-I. However, in terms of the robustness, IAD-I generally achieves the comparable or even better than IAD-II under both AT and TRADES.

## 5.2 EVALUATION ON *Tiny-ImageNet* DATASET

**Experiment Setup.** In this part, we evaluate these methods on a more challenging *Tiny-ImageNet* dataset. For these adversarially pre-trained models, we follow the settings of (Chen et al., 2021) to train AT and TRADES. To be specific, we train *PreActive-ResNet-18* using SGD with 0.9 momentum

Figure 4: Analysis about using different $\beta$ and warming-up periods in IAD: *Left*, the values of $\alpha$ (Eq. 4) under different $\beta$; *Middle*, Natural and PGD-20 accuracy of teacher model using different $\beta$; *Right*, Natural and PGD-20 accuracy of the teacher model with different warming-up periods.

for 100 epochs. The initial learning rate is 0.1 divided by 10 at Epoch 50 and 80 respectively, and the weight decay=0.0005. For distillation baselines, we keep most settings the same as Section 5.1. For ARD and IAD-I, here we set its $\lambda = 0.9$ to deal with the complex task following (Goldblum et al., 2020). And for both IAD-I and IAD-II, we use $\lambda = 0.1$, $\beta = 0.1$ and 10 warming-up epochs.

**Results.** We report the results in Table 4. Overall, our IAD-I or IAD-II can still achieve better model robustness than other methods. Specifically, on *Tiny-ImageNet*, IAD-II can approximately improve both the natural accuracy and the robust accuracy compared to IAD-I and other baselines.

## 5.3 ABLATION STUDIES

To give a comprehensive understanding of our proposed IAD method, we have conducted a series of experiments (in Appendix A), including the ablation study on using different $\gamma$ related to student introspection, different $\tau$ related to adversarial distillation and the loss terms of IAD-II as well as the comparison of the computational cost. In the following, we only study the effect of $\beta$ and warming-up periods for our student introspection. More ablation study can be found in the Appendix.

**Experiment Setup.** To understand the effects of different $\beta$ and different warming-up periods on CIFAR-10 dataset, we conduct the ablation study in this part. Specifically, we choose the ResNet-18 as the backbone model, and keep the experimental settings the same as Section 5.1. In the first experiments, we set no warming-up period and study the effect of using different $\beta$. Then, in the second experiments, we set $\beta = 0.1$ and investigate different warming-up periods.

**Results.** We report part of the results on IAD-I method in Figure 4. The complete results with other evaluation metrics as well as that on IAD-II method can be found in Appendix A.1. In Figure 4, we first visualize the values of the $\alpha$ using different $\beta$ in the left panel, which shows the proportion of the teacher guidance and student introspection in adversarial distillation. The bigger the beta corresponds to a larger proportion of the student introspection. In the middle panel, we plot the natural and PGD-20 accuracy of the student models distilled by different $\beta$. We note that the PGD-20 accuracy is improved when the student model trusts itself more with the larger $\beta$ value. However, the natural accuracy is decreasing along with the increasing of the $\beta$ value. Similarly, we adjust the length of warming-up periods and check the natural and PGD-20 accuracy in the right panel of Figure 4. We find that setting the student model partially trust itself at the beginning of the training process leads to inadequate robustness improvements. An appropriate warming-up period at the early stage can improve the student model performance on the adversarial examples.

## 6 CONCLUSION

In this paper, we study distillation from adversarially pre-trained models. We take a closer look at adversarial distillation and discover that the guidance of teacher model is progressively unreliable by considering the robustness. Hence, we explore the construction of reliable guidance in adversarial distillation and propose a method for distillation from unreliable teacher models, i.e., Introspective Adversarial Distillation. Our methods encourages the student model partially instead of fully trust the guidance of the teacher model and gradually trust its self-introspection more to improve robustness.

## 7 ACKNOWLEDGEMENT

JNZ and BH were supported by the RGC Early Career Scheme No. 22200720, NSFC Young Scientists Fund No. 62006202 and HKBU CSD Departmental Incentive Grant. JCY and HXY was supported by NSFC No. U20A20222. JFZ was supported by JST, ACT-X Grant Number JPMJAX21AF. TLL was supported by Australian Research Council Projects DE-190101473 and DP-220102121. JLX was supported by RGC Grant C6030-18GF.

## 8 ETHICS STATEMENT

This paper does not raise any ethics concerns. This study does not involve any human subjects, practices to data set releases, potentially harmful insights, methodologies and applications, potential conflicts of interest and sponsorship, discrimination/bias/fairness concerns, privacy and security issues, legal compliance, and research integrity issues.

## 9 REPRODUCIBILITY STATEMENT

To ensure the reproducibility of experimental results, our code is available at `https://github.com/ZFancy/IAD`.

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

APPENDIX

## A  COMPREHENSIVE EXPERIMENTS

### A.1  COMPLETE RESULTS OF ABLATION STUDIES.

In this section, we report the complete results of our ablation studies in Tables 5 (about $\beta$) and 6 (about warming-up periods).

Table 5: Test accuracy (%) of IAD-I and IAD-II using different $\beta$ with ResNet-18 on CIFAR-10.

|  | $\beta$ | Natural | FGSM | PGD-20 | $CW_\infty$ | AA |
|---|---|---|---|---|---|---|
|  | 0.01 | **82.76%** | 63.58% | 51.49% | 50.16% | 48.26% |
|  | 0.05 | 82.72% | **64.09%** | 51.51% | 50.43% | **48.73%** |
| IAD-I | 0.1 | 82.70% | 63.57% | 51.75% | **50.51%** | 48.53% |
|  | 0.5 | 80.66% | 62.79% | 52.36% | 50.19% | 48.43% |
|  | 1.0 | 80.79% | 63.55% | **52.64%** | 50.40% | 48.61% |
|  | 0.01 | **83.25%** | **64.46%** | 51.59% | 50.62% | 48.39% |
|  | 0.05 | 83.13% | 63.76% | 52.23% | 50.81% | 48.75% |
| IAD-II | 0.1 | 83.07% | 64.12% | 52.07% | 50.63% | 48.36% |
|  | 0.5 | 82.08% | 63.44% | 52.53% | **50.86%** | **49.02%** |
|  | 1.0 | 81.54% | 63.82% | **52.72%** | 50.76% | 48.75% |

Table 6: Test accuracy (%) of IAD-I and IAD-II using different warming-up periods with ResNet-18 on CIFAR-10.

|  | warming-up | Natural | FGSM | PGD-20 | $CW_\infty$ | AA |
|---|---|---|---|---|---|---|
|  | 0 epochs | 82.70% | 63.57% | 51.75% | 50.51% | 48.53% |
|  | 20 epochs | 82.75% | 63.82% | 51.50% | 50.44% | 48.37% |
| IAD-I | 40 epochs | **82.82%** | **63.96%** | 51.71% | 50.67% | 48.44% |
|  | 60 epochs | 82.09% | 63.20% | **52.14%** | **50.74%** | **48.66%** |
|  | 80 epochs | 82.17% | 63.40% | 52.03% | 50.59% | 48.57% |
|  | 0 epochs | 83.07% | 64.12% | 52.07% | 50.63% | 48.36% |
|  | 20 epochs | **84.22%** | **64.07%** | 51.60% | 50.38% | 48.47% |
| IAD-II | 40 epochs | 83.21% | 63.54% | 51.85% | 50.67% | **48.58%** |
|  | 60 epochs | 83.18% | 63.97% | **52.08%** | **50.68%** | 48.51% |
|  | 80 epochs | 83.07% | 63.92% | 51.96% | 50.30% | 48.09% |

In Table 5, we can see that the natural and FGSM accuracy will decrease, and the robust accuracy (PGD-20, CW∞, AA) will increase with the rise of $\beta$. In Table 6, we adjust the length of warming-up periods. We can see that letting the student network partially trust itself at the beginning of the training process would result in inadequate robustness improvements. In summary, we can find that there is a trade-off in Table 5 to achieve both larger natural accuracy and larger robustness in adversarial distillation, which is also similar to the standard adversarial training. We can slightly sacrifice the robustness (adjust the $\beta$ or adjust the warming-up periods) to acquire a better natural accuracy and FGSM.

### A.2  FULLY TRUST OR PARTIALLY TRUST THE TEACHER GUIDANCE

In this section, we compare the method with fully trusted or partially trusted teacher guidance. Regarding the variant of $AKD^2$ that replaces the second term in $AKD^2$ by our "partially trust" KL term (but without the introspection term), we find the robust improvement in Table 7.

Table 7: Comparison between AKD$^2$ and a "partially trust" variant with ResNet-18 on CIFAR-10.

|  | Natural | FGSM | PGD-20 | CW$_\infty$ |
|---|---|---|---|---|
| AKD$^2$ | **83.52%** | 63.91% | 51.36% | 50.36% |
| "partially trust" | 83.37% | **63.95%** | **51.49%** | **50.40%** |

In the "partially trust" variant of AKD$^2$, we just down weight the part of teacher guidance which has wrong prediction results with the hard labels. The results show that fit this part "unreliable" teacher guidance may improve the natural performance but consistently drop the robust accuracy.

In the following, we also conduct a range of experiments to compare IAD-I and its variant IAD-I-Down that applys our downweighting on $\ell_{kl}(S(\tilde{x}|\tau)||T(x|\tau))$ rather than a constant 1.0 like ARD. According to the results, we can see that IAD-I consistently achieves better robustness while sacrifices a little natural accuracy. Hence, we will choose the constant 1.0 on $\ell_{kl}(S(\tilde{x}|\tau)||T(x|\tau))$ for ARD in the main part of our experiments on those benchmark datasets.

Table 8: Comparison between IAD-I and IAD-I-Down. IAD-I-Down is a variant of IAD-I, where the weight on $\ell_{kl}(S(\tilde{x}|\tau)||T(x|\tau))$ is downweighted with $\alpha$ rather than a constant 1.0 like ARD.

| Dataset | Method | Natural | FGSM | PGD-20 | CW$_\infty$ |
|---|---|---|---|---|---|
| CIFAR-10 | IAD-I | 82.09% | 63.20% | **52.14%** | **50.74%** |
|  | IAD-I-Down | **83.33%** | **63.90%** | 51.77% | 50.63% |
| CIFAR-100 | IAD-I | 55.53% | **35.69%** | **27.40%** | **25.80%** |
|  | IAD-I-Down | **55.88%** | 35.68% | 27.32% | 25.60% |
| Tiny-ImageNet | IAD-I | 46.14% | **29.38%** | **23.40%** | **20.74%** |
|  | IAD-I-Down | **46.66%** | 29.30% | 22.68% | 19.64% |

## A.3 THE EFFECTS OF $\gamma$ FOR STUDENT INTROSPECTION

In this section, we check the effects of the Student Introspection via adjusting the $\gamma$ for our IAD methods. Here, we also use the constant $\gamma$ in the experiments. We find that using the constant $\gamma$ can further boost the performance on model robustness, but there is another problem that the natural accuracy will sacrifice a little bit more than our previous dynamical design, i.e., $\gamma = 1 - \alpha$.

Table 9: Test accuracy (%) of IAD-I and IAD-II using different coefficients for Student Introspection with ResNet-18 on CIFAR-10.

|  | coefficient | Natural | FGSM | PGD-20 | CW$_\infty$ |
|---|---|---|---|---|---|
| IAD-I | 1 - $\alpha$ | **82.09%** | 63.20% | 52.14% | 50.74% |
|  | 0.5 | 81.46% | **63.61%** | 52.77% | 50.67% |
|  | 1.0 | 80.04% | 63.52% | 54.14% | 50.88% |
|  | 2.0 | 77.59% | 62.86% | 55.64% | 50.72% |
|  | 4.0 | 77.34% | 62.68% | **55.78%** | **51.29%** |
| IAD-II | 1 - $\alpha$ | **83.21%** | 63.54% | 51.85% | 50.67% |
|  | 0.5 | 82.76% | 63.91% | 52.50% | 50.67% |
|  | 1.0 | 81.33% | 63.48% | 53.32% | 50.84% |
|  | 2.0 | 80.95% | 63.90% | 54.02% | 51.13% |
|  | 4.0 | 80.43% | **64.05%** | **55.27%** | **51.97%** |

According to the results in Table 9, we can hardly to find an optimal coefficient for the student introspection term to achieve both the best natural accuracy and the best robustness. However,

there is one trend that increasing the coefficient will gain more robustness with losing more natural accuracy. With the hyper-parameter $\gamma$, we may flexibly instantiated by some constants or some strategic schedules to pursue the robustness or the natural accuracy.

## A.4 THE EFFECTS OF $\tau$ FOR ADVERSARIAL DISTILLATION

In this section, we check the effects of $\tau$ for adversarial distillation under the same training configurations and also for TRADES. We have listed the results of TRADES in Tabel 10, and the results of IAD-I and ARD in Table 11.

Table 10: Test accuracy (%) of TRADES with different $\tau$ with ResNet-18 on CIFAR-10.

| $\tau$ | 0.1 | 1 | 20 | 60 | 100 |
|---|---|---|---|---|---|
| Natural | **87.14%** | 81.26% | 84.57% | 83.95% | 84.43% |
| PGD-20 | 43.26% | **52.98%** | 51.18% | 51.17% | 50.73% |

As for TRADES, we think it may not need the temperature $\tau$ which is especially designed for distillation, since the KL term in TRADES is aims to encourage the output of the model on adversarial data to being the similar with that on natural data. Intuitively, it encourage the output to be more stable which leads to better robust performance. As a result, based on the test accuracy in Table 10, $\tau = 1$ maybe the best for TRADES to achieve a better robustness and enlarging (or decreasing) $\tau$ can disturb the original output logits and may result in lower robustness (i.e., $\tau = 0.1$: PGD-20 43.26% ).

Table 11: Test accuracy (%) of IAD-I and ARD with different $\tau$ with ResNet-18 on CIFAR-10.

| | $\tau$ | 0.1 | 1 | 20 | 60 | 100 |
|---|---|---|---|---|---|---|
| IAD-I | Natural | 82.76% | **82.09%** | 82.22% | 82.12% | 81.95% |
| | PGD-20 | **55.71%** | 52.14% | 50.74% | 50.83% | 51.02% |
| ARD | Natural | 83.11% | 83.13% | **83.18%** | 82.39% | 82.89% |
| | PGD-20 | **55.81%** | 51.36% | 50.44% | 50.37% | 50.15% |

According to the comparison in Table 11, with the proper $\tau$ (*e.g.,* $\tau \geq 1$), IAD-I can achieve the comparable natural accuracy and better robustness. Note that in the experiments of IAD-I, we keep the $\tau = 1$ for the student introspection term according to the results of TRADES in Table 10.

## A.5 COMPARISON WITH TRADES AND ARD

In this section, we compared the performance of TRADES, ARD and our IAD-I with different weights in their each loss terms. We summarized the results in Table 12.

Table 12: Test accuracy (%) of using different weights for the loss terms in TRADES and ARD.

| Method | Label $\ell_{CE}(S(x), y)$ | Self $\ell_{KL}(S(\tilde{x}|\tau)||S(x|\tau))$ | Teacher $\ell_{KL}(S(\tilde{x}|\tau)||T_{at}(\tilde{x}|\tau))$ | Accuracy Natural | PGD-20 |
|---|---|---|---|---|---|
| TRADES | 1 | 6 | 0 | 81.26% | 52.98% |
| ARD | 0 | 0 | 1 | 83.13% | 51.36% |
| - | 1 | 0 | 1 | **85.94%** | 48.58% |
| - | 0 | 6 | 1 | 74.95% | **55.89%** |
| - | 1 | 6 | 1 | 79.73% | 54.07% |
| - | 0 | 1-$\alpha$ | $\alpha$ | 83.33% | 51.77% |
| IAD-I | 0 | 1 | $\alpha$ | 82.09% | 52.14% |

According to the results, IAD-I can approximately achieve the better robust accuracy than ARD, and the better natural accuracy than TRADES with less drop on PGD-20 accuracy. With varying the

hyper-parameters on the loss terms, we also find that natural accuracy can reach to $85.94\%$ at a loss of robustness to $48.58\%$ and robustness can reach to $55.89\%$ at a loss of natural accuracy to $74.95\%$. In summary, through adjust those weight, our IAD-I can flexibly make a good balance among them by considering both the natural accuracy and the robustness.

## A.6    ABLATION STUDY ON THE LOSS TERMS OF IAD-II

In this section, we conduct the ablation study in Table 13 about the four loss terms in IAD-II which is based on $AKD^2$.

Table 13: The ablation study about the different terms in IAD-II with ResNet-18 on CIFAR-10.

| Label | Self | Teacher | | Accuracy | |
| --- | --- | --- | --- | --- | --- |
| $\ell_{CE}(S(\tilde{x}),y)$ | $\ell_{KL}(S(\tilde{x}|\tau)\|S(x|\tau))$ | $\ell_{KL}(S(\tilde{x}|\tau)\|T_{at}(\tilde{x}|\tau))$ | $\ell_{KL}(S(\tilde{x}|\tau)\|T_{st}(\tilde{x}|\tau))$ | Natural | PGD-20 |
| ✓ | | | ✓ | **84.30%** | 50.67% |
| ✓ | | ✓ | | 82.66% | 51.72% |
| ✓ | | ✓ | ✓ | 83.52% | 51.36% |
| ✓ | ✓ | ✓ | ✓ | 83.21% | **51.85%** |

Specifically, according to the results, we find $\ell_{KL}(S(\tilde{x}|\tau)\|T_{st}(\tilde{x}|\tau))$ can help model gain more natural accuracy, while $\ell_{KL}(S(\tilde{x}|\tau)\|T_{at}(\tilde{x}|\tau))$ can help model gain more robustness. $AKD^2$ achieves a good balance between two aspects by combining above two terms, and IAD-II further boosts the model robustness by incorporating the self-introspection term with less drop in natural performance.

## A.7    COMPUTATIONAL COST COMPARISON

In this section, we check and compare the computational cost of ARD, $AKD^2$, IAD-I and IAD-II in terms of the averaged training time in each epoch, as well as the GPU Memory-Usage. The detailed results are summarized in Table 14.

Table 14: Time and Memory cost of adversarial distillation methods with ResNet-18 on CIFAR-10.

| Method | Time (Avg. Epoch) | GPU Memory-Usage |
| --- | --- | --- |
| ARD | 156.12s | 3253MiB |
| IAD-I | 182.01s | 3977MiB |
| $AKD^2$ | 147.10s | 2397MiB |
| IAD-II | 167.23s | 3249MiB |

According to the results, IAD-I and IAD-II correspondingly consume a bit more time and memory than ARD and $AKD^2$ due to the additional self-introspection. Besides, an interesting phenomenon is that ARD has less terms than $AKD^2$, but consumes more time and GPU memory. This is because ARD has to deal with both $x$ and $\tilde{x}$, while $AKD^2$ only needs to deal with $\tilde{x}$.

