# OpenReview forum: "Reliable Adversarial Distillation with Unreliable Teachers"
_ICLR.cc/2022/Conference — ICLR 2022 Poster_

### Official Review · Reviewer_Q98Y · 2021-11-01

**Correctness:** 4
**Technical Novelty And Significance:** 4
**Empirical Novelty And Significance:** 4
**Recommendation:** 6
**Confidence:** 4

**Main Review:**

Pros:
+ The research direction is promising and more practical than adversarial training in the real world. As I demonstrated in the summary part, in many ML-driven scenarios, users might not want to retrain their big deep networks using adversarial training due to huge computational cost. Thus, it is, IMHO, urgent to consider to distil the robustness of a robust model.
+ Although the high-level idea is mentioned in previous papers, this paper considers this problem in detail and find a failure case of previous papers. By considering this failure case, this paper proposes a novel framework to complete this task, which is novel and significant to the development of the field.
+ This paper is well-written and easy to follow. Experiments are enough to support the claims made in this paper. A plus should be that TRADES and AT are both considered in the experiments, which verifies that the proposed framework is suitable for existing adversarial training methods.

Cons:
- Since the student model is different from the teacher model, it is expected that adversarial data generated by the student model is different from the data used to train the teacher model. This kind of performance drop also happens in other fields, such as domain adaptation, even adversarial machine learning itself. Thus, it is not clear what the point of Figure 1(a) is. I am not sure if 1(a) is necessary. It is more like the transferability of adversarial attacks.
- Based on the above drawback, it is also unclear if there are other points that make the DISTILLATION fails (except for the distributional discrepancy).
- The formal problem setting is missing. It is better to provide the formal problem setting to make readers understand the problem clearly.
- What is the difference between S(\circ|\tau) and S(\circ)? I did not see any definition of S(\circ|\tau). How does \tau effect on S(\circ|\tau)?
- I would like to see how \tau effects the performance of the proposed method. If we remove \tau, then the “Student Introspection” term is actually from TRADES. Meanwhile, how does \tau effect the performance of TRADES? Are there any papers discussing this point?
- From the experimental results, IAD performs much better than ARD (IMHO, the most direct baseline to IAD), which is very good. I would like to see how \tau effects the performance of IAD compared to ARD (giving the same \tau for ARD and IAD).
- The computational-cost comparison is missing, which is important for distillation-based method.
- Figure 1(b) is a clear figure, but I cannot see the advantages of your method in this figure. In other words, it is unclear why your method can improve the performance when reading 1(b).


**Summary Of The Paper:**

Adversarial training has been developed for many years, and it aims to provide reliable classifiers in the era of deep learning. However, in many ML-driven scenarios, users might not want to retrain their big deep networks using adversarial training. The main reason is that training a reliable/adversarial-robust classifier will cost many computational resources. To address this issue, some methods are proposed to distil the robustness from adversarially pre-trained models, which is more practical than the direct adversarial training. This is also the point this paper focuses on. In my humble opinion, this is a very promising research direction and should be paid more attentions in future.
Compared to existed works, this paper argues that the adversarial training data of the student model and that of the teacher model are egocentric (respectively generated by themselves) and becoming more adversarial challenging during training, which causes failure of existing works. To address this issue, this paper proposes an Introspective Adversarial Distillation (IAD) to effectively utilize the knowledge from an adversarially pre-trained teacher model. Extensive experiments are conducted on CIFAR-10/CIFAR-100 and the more challenging Tiny-ImageNet datasets to evaluate the efficiency of our IAD.


**Summary Of The Review:**

In general, considering the significance of the researched problem, this paper can be accepted by the ICLR2022. However, some points should be clarified and strengthened in the revision. I would like to strongly support this paper if my concerns can be fully addressed.

---

> ### Author Response · Authors · 2021-11-17
> **Response to Reviewer Q98Y: Part 1**
>
> > **Q1:**  Since the student model is different from the teacher model, it is expected that adversarial data generated by the student model is different from the data used to train the teacher model. This kind of performance drop also happens in other fields, such as domain adaptation, even adversarial machine learning itself. Thus, it is not clear what the point of Figure 1(a) is. I am not sure if 1(a) is necessary. It is more like the transferability of adversarial attacks.
>
> **A1:** Except comparison with the ordinary distillation, the main goal of Figure 1(a) is to characterize the teacher network's degeneration trend on adversarial images along with the adversarial training. Like the phenomenon in domain adaptation, we might envision the performance drop due to the distribution discrepancy, but here we might not understand how it changes in this training process. Specifically, in adversarial distillation, the reliability of the teacher network on adversarial images progressively becomes devalued. We will improve this part to avoid potential confusion.
>
> > **Q2:**  Based on the above drawback, it is also unclear if there are other points that make the DISTILLATION fails (except for the distributional discrepancy).
>
> **A2:** Thanks for your question! Actually, there are other points that might make the distillation fail. For example, if we choose a smaller network as the student, it might not well mimic the teacher even its soft labels on adversarial images are reliable. In this case, there will be two latent factors that perplex us about this phenomenon. To remove some potential confounders, we follow the convention of ARD and AKD$^2$ that utilizes a student network same to the teacher. However, we really would like to span our explore on this point by conducting more experiments in the future version.
>
> > **Q3:**  The formal problem setting is missing. It is better to provide the formal problem setting to make readers understand the problem clearly.
>
> **A3:** Thanks for the advice. We will add a formal definition in the draft. Specifically, the informal snapshot of adversarial distillation is $E_{\tilde{x}\in \mathcal{B}[x]}\left[\ell_{kl}(S(\tilde{x}|\tau)||T(\tilde{x}|\tau))\right]$, where $T(\tilde{x}|\tau)$ is not a constant soft supervision along with the adversarial training and affected by the adversarial data generated by the dynamically evolved student network.
>
> > **Q4:** What is the difference between $S(\cdot|\tau)$ and $S(\cdot)$? I did not see any definition of $S(\cdot|\tau)$. How does $\tau$ effect on $S(\cdot|\tau)$?
>
> **A4:**  $S(\cdot|\tau)$ is a Softmax operator with the temperature $\tau$, *e.g.,* $S(x_k|\tau)=\frac{\exp(x_k/\tau)}{\sum_{k'} \exp(x_{k'}/\tau)}$, which is usually used in knowledge distillation. And $S(\cdot)$ is the conventional Softmax with the temperature $\tau=1$. We will include the definition below Eq.(3) for clarity in the submission.
>
> > **Q5 \& Q6:** I would like to see how $\tau$ effects the performance of the proposed method. If we remove $\tau$, then the “Student Introspection” term is actually from TRADES. Meanwhile, how does $\tau$ effect the performance of TRADES? Are there any papers discussing this point? From the experimental results, IAD performs much better than ARD (IMHO, the most direct baseline to IAD), which is very good. I would like to see how $\tau$ effects the performance of IAD compared to ARD (giving the same $\tau$ for ARD and IAD).
>
> **A5 \& A6:** We have listed the results of IAD and ARD under different $\tau$ in the following table (Table 1).
>
> **Table 1.** Test accuracy of IAD and ARD with different $\tau$ with ResNet-18 on CIFAR-10.
>
> |      |   $\tau$   |   0.1   |   1    |   20   |   60   |   100   |
> | :--: | :--------: | :-----: | :----: | :----: | :----: | :-----: |
> | IAD  |  Natural   | 82.86%  | 83.33% | 82.27% | 82.28% | 82.40%  |
> | IAD  |   PGD-20   | 55.97%  | 51.77% | 50.74% | 50.65% | 50.60%  |
> |      | **$\tau$** | **0.1** | **1**  | **20** | **60** | **100** |
> | ARD  |  Natural   | 83.11%  | 83.13% | 83.18% | 82.39% | 82.89%  |
> | ARD  |   PGD-20   | 55.81%  | 51.36% | 50.44% | 50.37% | 50.15%  |
>
> According to the comparison, with the proper $\tau$ (*e.g.,* $\tau\geq 1$), IAD can achieve comparable or better natural accuracy and robustness. Regarding the $\tau$ *w.r.t* TRADES, to our knowledge, we have not found any discussion. We conduct the experiments to verify its effect on TRADES as follows (Table 2), and will include these results into our Appendix.
>
> **Table 2.** Test accuracy of TRADES with different $\tau$ with ResNet-18 on CIFAR-10.
>
> | $\tau$  |  0.1   |   1    |   20   |   60   |  100   |
> | :-----: | :----: | :----: | :----: | :----: | :----: |
> | Natural | 87.14% | 81.26% | 84.57% | 83.95% | 84.43% |
> | PGD-20  | 43.26% | 52.98% | 51.18% | 51.17% | 50.73% |

---

> ### Author Response · Authors · 2021-11-17
> **Response to Reviewer Q98Y: Part 2**
>
> > **Q7:** The computational-cost comparison is missing, which is important for distillation-based method.
>
> **A7:** The computational cost is summarized in the following table (Table 3) and will be added to the Appendix. IAD-I (*i.e.,* original IAD, on basis of ARD) and IAD-II (*i.e.,* IAD + AKD$^2$, our method IAD on basis of the AKD$^2$) correspondingly consume a bit more time and memory than ARD and AKD$^2$ due to the additional self-introspection. Besides, an interesting phenomenon is that ARD has fewer terms than AKD$^2$, but consumes more time and GPU memory. This is because ARD has to deal with both $x$ and $\tilde{x}$, while AKD$^2$ only needs to deal with $x$.
>
> **Table 3.** Time and Memory cost of adversarial distillation methods.
>
> | Method  | Time (Avg. Epoch) | GPU Memory-Usage |
> | :-----: | :---------------: | :--------------: |
> |   ARD   |      156.12s      |     3253MiB      |
> |  IAD-I  |      182.01s      |     3977MiB      |
> | AKD$^2$ |      147.10s      |     2397MiB      |
> | IAD-II  |      167.23s      |     3249MiB      |
>
> > **Q8:** Figure 1(b) is a clear figure, but I cannot see the advantages of your method in this figure. In other words, it is unclear why your method can improve the performance when reading 1(b).
>
> **A8:**  Thank you. We will improve the caption by kindly referring to the evidence about the student introspection in Figure 3. Besides, we will improve the explanation in the corresponding paragraph of the Introduction.

---

### Official Review · Reviewer_xcEE · 2021-11-01

**Correctness:** 4
**Technical Novelty And Significance:** 4
**Empirical Novelty And Significance:** 3
**Recommendation:** 8
**Confidence:** 4

**Main Review:**

In summary, the discovery of the specific degeneration in adversarial distillation is interesting, and the authors design an elaborate loss that takes such a drawback in the adversarial distillation into account. A range of experiments on different network architectures, adversarial training backbones (AT or TRADES) and three widely-used datasets demonstrate its superiority in improving the model robustness. Besides, the ablation study regarding the annealing temperature and the warming up has been conducted to provide the view on its working mechanism.

However, there are still some concerns about this work.
Major Concerns
1)  In Section 3.2, the authors analyze the adversarial data by partitioning them into three groups. Is there the fourth group where the teacher model has wrong predictions on natural data but right prediction on its adversarial counterpart? If it almost does not exist, it will be complete that the authors at least mention this in the paragraph. Besides, for the third group, it is not absolutely consistent with the case of x_2 where it also requires the wrong prediction of the student model.
2) In Section 4, Eq.(4) can approximately reflect the confidence on the adversarial data. But have the authors considered the difference between group 2 and group 3 in Eq.(4), since they may have different scale and corresponds to different cases.
3) In fact, the proposed Introspective Adversarial Distillation can degenerate to TRADES and ARD by setting some extreme hyperparameters. It will be better to conduct an experiment to plot a accuracy (natural acc and robust acc) surface with varying hyperparameters that includes these three methods, which helps us to better understand the position your method.

Minor Concerns
1) Table 1 is messy. The authors could list each distance term as a column name and compare baselines by considering it or not.
2) The experiments of IAD + AKD^2 is not well explained. I guess it is to show how IAD can help improve the natural accuracy but the authors have not clarified this.


**Summary Of The Paper:**

This paper studies a specific distillation scenario, adversarial distillation, to improve the model robustness. Different from the constant sort supervision in the ordinary distillation, the teacher model will become progressively unreliable along with training, since the adversarial data are dynamically searched by the student model and might not be well identified by the teacher model. Therefore, the authors introduce an introspective adversarial distillation, which considers to bootstrap the learning from both the teacher model and itself. A range of experiments demonstrate its superiority on improving the model robustness.

**Summary Of The Review:**

This paper investigates an interesting problem about adversarial distillation, which has some distinct with the ordinary distillation. The authors provides an introspective adversarial distillation to solve the problem. Despite simple, it shows consistent improvement on the model robustness while does not sacrifices the natural accuracy too much. However, the writing about the motivation and the loss designed for each group as well as the experimental parts are not very clear and need to be further improved.

---

> ### Author Response · Authors · 2021-11-17
> **Response to Reviewer xcEE: Part 1**
>
> > **Q1:**  In Section 3.2, the authors analyze the adversarial data by partitioning them into three groups. Is there the fourth group where the teacher model has wrong predictions on natural data but right prediction on its adversarial counterpart? If it almost does not exist, it will be complete that the authors at least mention this in the paragraph. Besides, for the third group, it is not absolutely consistent with the case of $x_2$ where it also requires the wrong prediction of the student model.
>
> **A1:**  Considering the generation process of the adversarial data, *i.e.,* $\tilde{x}^* = \arg\max_{\tilde{x}\in \mathcal{B}_\epsilon(x)} \ell(f(\tilde{x}), y)$, there are actually three groups only. That is, when $T(x)=y$, $\tilde{x}^*$ belongs to two cases where the adversarial data can successfully confuse the model or not. But when $T(x)\neq y$, there is only one case that $\tilde{x}^*$ makes the prediction fail. This is because the optimization to generate the data is towards at maximizing the loss on basis of natural data. Once the original prediction is wrong, *i.e.,*, $T(x)\neq y$, the generation of $\tilde{x}^*$ only make the prediction worse. Thus, it is not possible that $T(x)\neq y$ and simultaneously $T(\tilde{x}^*)=y$. We will follow the advice to discuss this special case in the paper.
>
> > **Q2:**  In Section 4, Eq.(4) can approximately reflect the confidence on the adversarial data. But have the authors considered the difference between group 2 and group 3 in Eq.(4), since they may have different scale and corresponds to different cases.
>
> **A2:**  We have considered this difference, but we find the scale is relatively continuous by controlling the annealing parameter $\beta$ as shown in the left panel of Figure 4. Thus, we adopt one unified confidence term to reflect the importance of the teacher network on the adversarial data as in Eq.(4).
>
> > **Q3:**  In fact, the proposed Introspective Adversarial Distillation can degenerate to TRADES and ARD by setting some extreme hyperparameters. It will be better to conduct an experiment to plot a accuracy (natural acc and robust acc) surface with varying hyperparameters that includes these three methods, which helps us to better understand the position your method.
>
> **A3:**  The corresponding experiments are summarized in the following table. According to the results, IAD achieves the comparable natural accuracy with ARD and the comparable robustness with TRADES. But with varying the hyperparameters, natural accuracy can reach to $85.94\%$ at a loss of robustness to $48.58\%$ and robustness can reach to $55.89\%$ at a loss of natural accuracy to $74.95\%$. IAD makes a good balance among them by considering both the natural accuracy and the robustness.
>
> **Table 1.** Test accuracy of using different weights for the loss terms in TRADES and ARD in CIFAR-10.
>
> |            |        Label         |                    Self                    |                         Teacher                         |             |            |
> | :--------: | :------------------: | :----------------------------------------: | :-----------------------------------------------------: | :---------: | :--------: |
> | **Method** | $\ell_{CE}(S(x), y)$ | $\ell_{KL}(S(\tilde{x}\|\tau)\|\| S(x\|\tau))$ | $\ell_{KL}(S(\tilde{x}\|\tau)\|\| T_{at}(\tilde{x}\|\tau))$ | **Natural** | **PGD-20** |
> |   TRADES   |          1           |                     6                      |                            0                            |   81.26%    |   52.98%   |
> |    ARD     |          0           |                     0                      |                            1                            |   83.13%    |   51.36%   |
> |     -      |          1           |                     0                      |                            1                            | **85.94%**  |   48.58%   |
> |     -      |          0           |                     6                      |                            1                            |   74.95%    | **55.89%** |
> |     -      |          1           |                     6                      |                            1                            |   79.73%    |   54.07%   |
> |    IAD     |          0           |                 1-$\alpha$                 |                        $\alpha$                         |   83.33%    |   51.77%   |

---

> ### Author Response · Authors · 2021-11-17
> **Response to Reviewer xcEE: Part 2**
>
> > **Q4:** Table 1 is messy. The authors could list each distance term as a column name and compare baselines by considering it or not. The experiments of IAD + AKD$^2$ is not well explained. I guess it is to show how IAD can help improve the natural accuracy but the authors have not clarified this.
>
> **A4:**  Thank you for the suggestion. We revised Table 1 as suggested (will be revised in the updated version), and used IAD-II as the surrogate of the combination of IAD+AKD$^2$ by following reviewer 2bpg. We will update it and clear our clarity in the updated version.
>
> **Table 2.** Loss comparison. The $\checkmark$ means that the term is considered in the model objective.
>
> | Knowledge |        Label         |            Label             |                    Self                    |                     Teacher                     |                         Teacher                         |                         Teacher                         |
> | :-------: | :------------------: | :--------------------------: | :----------------------------------------: | :---------------------------------------------: | :-----------------------------------------------------: | :-----------------------------------------------------: |
> | Loss Term | $\ell_{CE}(S(x), y)$ | $\ell_{CE}(S(\tilde{x}), y)$ | $\ell_{KL}(S(\tilde{x}\|\tau)\|\| S(x\|\tau))$ | $\ell_{KL}(S(\tilde{x}\|\tau)\|\| T_{at}(x\|\tau))$ | $\ell_{KL}(S(\tilde{x}\|\tau)\|\| T_{at}(\tilde{x}\|\tau))$ | $\ell_{KL}(S(\tilde{x}\|\tau)\|\| T_{st}(\tilde{x}\|\tau))$ |
> |    AT     |                      |         $\checkmark$         |                                            |                                                 |                                                         |                                                         |
> |  TRADES   |     $\checkmark$     |                              |                $\checkmark$                |                                                 |                                                         |                                                         |
> |    ARD    |     $\checkmark$     |                              |                                            |                  $\checkmark$                   |                                                         |                                                         |
> |   IAD-I   |     $\checkmark$     |                              |                $\checkmark$                |                  $\checkmark$                   |                                                         |                                                         |
> |  AKD$^2$  |                      |         $\checkmark$         |                                            |                                                 |                      $\checkmark$                       |                      $\checkmark$                       |
> |  IAD-II   |                      |         $\checkmark$         |                $\checkmark$                |                                                 |                      $\checkmark$                       |                      $\checkmark$                       |

---

### Official Review · Reviewer_svED · 2021-11-02

**Correctness:** 3
**Technical Novelty And Significance:** 3
**Empirical Novelty And Significance:** 3
**Recommendation:** 6
**Confidence:** 4

**Main Review:**

Pros.

The overall idea is interesting and the proposed method seems simple and practical as well since we only need to adjust the annealing parameter adaptively. Besides, the paper is quite readable and experimental results seem promising.

Cons.

There are some issues to be addressed.

- The results of Natural and FGSM are not good. The tradeoff between Natural performance and robustness should be investigated more.

- In tiny-imagenet and cifar datasets, why IAD + AKD^2 is better than IAD for AA? This phenomenon should be explained more.

- The current evaluation is based on image datasets, like cifar and tiny-imagenet. However, is IAD still working for language datasets?

- It is good to discuss some theoretical justification for IAD, which will make IAD stronger in theory.

**Summary Of The Paper:**

In this paper, a new method called introspective adversarial distillation (IAD) is proposed for conventional adversarial training. Concretely, it targets the unreliable teacher case, where teacher is good at adversarial / natural data or none of it. Experimental results validate the effectiveness of the proposed method towards enhancing adversarial robustness.

**Summary Of The Review:**

Basically, this paper is well written and the proposed method seems promising. I thus would like to lean on the acceptance side.

---

> ### Author Response · Authors · 2021-11-17
> **Response to Reviewer svED**
>
> > **Q1:**  The results of Natural and FGSM are not good. The tradeoff between Natural performance and robustness should be investigated more.
>
> **A1:** Thank you for the suggestion. Actually, we have conducted some experiments to visualize this trade-off by controlling $\beta$. For clarity, we put the table of results from the Appendix in the following.
>
> **Table 1.** Test accuracy of IAD using different $\beta$ with ResNet-18 on CIFAR-10.
>
> | $\beta$ |  Natural   |    FGSM    |   PGD-20   |     CW     |     AA     |
> | :-----: | :--------: | :--------: | :--------: | :--------: | :--------: |
> |  0.01   | **83.16%** | **63.97%** |   51.28%   |   50.38%   |   48.20%   |
> |  0.05   |   82.82%   |   63.12%   |   51.51%   |   50.43%   |   48.19%   |
> |   0.1   |   82.32%   |   63.66%   |   51.70%   |   50.51%   |   48.30%   |
> |   0.5   |   80.39%   |   63.17%   |   52.62%   |   50.76%   |   49.01%   |
> |   1.0   |   78.61%   |   61.96%   | **52.81%** | **50.79%** | **49.21%** |
>
> From the results, we can find that it is a trade-off to achieve both larger natural accuracy and larger robustness with varying $\beta$, which is also similar to the standard adversarial training. We can slightly sacrifice the robustness to acquire a better natural accuracy and FGSM. More explanations about this trade-off will be added in the regular part of the submission.
>
> > **Q2:**  In tiny-imagenet and cifar datasets, why IAD + AKD$^2$ is better than IAD for AA? This phenomenon should be explained more.
>
> **A2:**  IAD + AKD$^2$ is our method IAD on basis of the AKD$^2$. For clarity, we term IAD + AKD$^2$ as IAD-II and the original IAD on basis of ARD as IAD-I. From the results in tiny-imagenet, IAD-II achieves comparable or better robustness than IAD-I. This might be because AKD$^2$ is generally a better starting point for IAD. But when the teacher network is trained by TRADES, there is a reverse phenomenon. We kindly refer the reviewer to the A1&Q1 of reviewer 2bpg for a more comprehensive comparison and explanation.
>
> > **Q3:**  The current evaluation is based on image datasets, like cifar and tiny-imagenet. However, is IAD still working for language datasets?
>
> **A3:** It is really a good question. We have to admit that there might be some challenges to implementing experiments on the language datasets. Specifically, compared to the pixel-wise perturbation in the images, it lacks some generally agreed ways to generate the adversarial sample for the textual data *e.g.,* the adversarial example of one sentence. Given that the research in this direction is still in the preliminary stage, we would like to complement the corresponding experiments in the future if some trials of adversarial training on the language datasets are persuasive.
>
> > **Q4:** It is good to discuss some theoretical justification for IAD, which will make IAD stronger in theory.
>
> **A4:** Thank you for the advice. We will try to provide a theoretical understanding about IAD based on the recent deep learning theory like [1] and improve the in-depth explanation in the submission.
>
> [1] Allen-Zhu, Zeyuan, and Yuanzhi Li. "Towards understanding ensemble, knowledge distillation and self-distillation in deep learning." arXiv preprint arXiv:2012.09816 (2020).

---

### Official Review · Reviewer_2bpg · 2021-11-03

**Correctness:** 3
**Technical Novelty And Significance:** 3
**Empirical Novelty And Significance:** 3
**Recommendation:** 6
**Confidence:** 3

**Main Review:**

Pros:
1. The paper is well written and easy to follow. The main claims and key observations are clearly stated.

2. The key observation is very inspiring: The (adversarially pretrained) teacher model's accuracy on adversarial images generated by the student model gradually drops. This provides solid foundation for the main claim of the paper: The student model shouldn't be fully convinced by the soft labels provided by the teacher model in adversarial distillation. This is something important that previous works overlooked.

3. AKD2+IDA does generally outperform previous methods, although the margin is not that significant. I suggest the authors to try the modification I mentioned in the second bullet in "Cons" if haven't already. It may enlarge the performance improvement based on my intuitions.

Cons:

As I mentioned before, I'm totally convinced with the intuition to partially trust the teacher model's soft label in adversarial distillation. Bellow are some minor concerns.

1. The observation in Figure 1 (a) and Figure 3 (left column) tells us that the teacher model's predictions on the **adversarial** images generated by the student model shouldn't be fully trusted. However, the previous method ARD uses the soft label generated by the teacher network on **clean** images. So, it is not proper to claim that ARD has issue because it fully trusted the teach model's unreliable soft labels. In fact, the soft labels used in ARD are generated by teacher on **clean** images and has been shown to be generally reliable by yourself in the blue curve in Figure 1 (a). With that said, ADK2 does use the soft-label on **adversarial** images. I think it is better to use ADK2 as the starting point to motivate your method, instead of from ARD as in Sec 3.1 of current version. An even more direct way to motivate IDA is to replace the second loss term (ie the "fully trust" KL term) in ADK2 to your "partially trust" KL term, while keeping other terms unchanged.

2. Most importantly, just like in ARD, IAD in Eq. (3) uses soft labels from teacher on **clean** images, instead of **adversarial** images. As I mentioned in bullet 1, the soft labels on **clean** images are generally trustworthy, as shown by yourself in the paper. So why does IAD partially trust it? It is never motivated, if I understand correctly.

3. To support your intuitions, the current design of IAD in Eq. (3) might not be the best choice. Specifically, the "Student Introspection" KL loss is weighted by (1-\alpha). This is not so intuitive for me. You have justified the "Teacher Guidance" is not always trustworthy so it is good to weight its loss term with \alpha. However, a trustworthy "Teacher Guidance" doesn't mean an untrustworthy "Student Introspection". So why do you down-weight "Student Introspection" KL loss when \alpha is large? In fact, the  "Student Introspection" KL loss term is just an annealed version of the smoothness loss term in TRADES. As shown in TRADES paper, it regularizes the smoothness of the model which helps robustness. In my view, based on your key motivation to partially trust the **teacher**, the **student** introspection loss term should have a constant weight and instead of dynamically down-weighted as in Eq (3).

4. There are four loss terms in the best performing method AKD2+IDA. It would be nice if ablation studies can be provided on all those loss terms.

Overall, I think the key observations in this paper is very inspiring. If proper modifications can be made to address the above issues, I think it would be a good paper.

**Summary Of The Paper:**

This paper proposes a new knowledge distillation (KD) method for adversarial training. The authors first observed that the soft-labels provided by the teacher gradually becomes less and less reliable during the adversarial training of student model. Based on this observation, they propose to partially trust the soft labels provided by the adversarily pretrained teacher.

**Summary Of The Review:**

Interesting observation and good intuition, but there is a noticeable gap between the designed method and their intuition. I agree the proposed IAD empirically works, but the intuitions provided in the paper to motivate and interpret IAD is problematic.

---

> ### Author Response · Authors · 2021-11-17
> **Response to Reviewer 2bpg: Part 1**
>
> > **Q1:** The observation ... I think it is better to use AKD2 as the starting point to motivate your method, instead of from ARD as in Sec 3.1 of current version. An even more direct way to motivate ...
>
> **A1:**  Thank you for the constructive suggestion. We have thoroughly validated the performance of IAD on basis of ARD (termed as IAD-I) and IAD on basis of AKD$^2$ (termed as IAD-II) in the following two tables (Tables 1 and 2), where the teacher network is trained by AT or TRADES. According to the first table, AKD$^2$ is actually a better starting point for IAD except for the case on CIFAR-100 with WideResNet-34-10. In the second table, however, it seems that IAD-II makes a trade-off between the natural accuracy and the robustness over IAD-I. Considering their distillation philosophy, *i.e.,* $\ell_{kl}(S(\tilde{x}|\tau)||T(x|\tau)))$ of IAD-I and $\ell_{kl}(S(\tilde{x}|\tau)||T(\tilde{x}|\tau)))$ of IAD-II, it might be up to which of $T(x|\tau)$ and $T(\tilde{x}|\tau)$ is more informative in adversarial distillation. With the reviewer's advice, we build an abstraction operator that can be instantiated by ARD or AKD$^2$, and combine it with the student introspection as a general IAD. Specifically, we will emphasize that AKD$^2$ is a better choice as the starting point. These contents will be updated in the revised submission.
>
> **Table 1.** Comparison between IAD-I (IAD on basis of ARD) and IAD-II (IAD on basis of AKD$^2$) when the teacher network is trained by AT.
>
> |    Dataset    |       Network       | Method |  Natural   |    FGSM    |   PGD-20   |     CW     |     AA     |
> | :-----------: | :-----------------: | :----: | :--------: | :--------: | :--------: | :--------: | :--------: |
> |   CIFAR-10    |      ResNet-18      | IAD-I  | **83.33%** | **63.90%** |   51.77%   |   50.63%   | **48.59%** |
> |   CIFAR-10    |      ResNet-18      | IAD-II |   83.21%   |   63.54%   | **51.85%** | **50.67%** |   48.58%   |
> |   CIFAR-10    |  WideResNet-34-10   | IAD-I  |   84.10%   |   66.24%   |   55.37%   |   54.20%   |   52.03%   |
> |   CIFAR-10    |  WideResNet-34-10   | IAD-II | **85.09%** | **66.54%** | **55.45%** | **54.63%** | **52.29%** |
> |   CIFAR-100   |      ResNet-18      | IAD-I  |   55.88%   |   35.68%   |   27.32%   |   25.60%   |   23.96%   |
> |   CIFAR-100   |      ResNet-18      | IAD-II | **57.26%** | **35.94%** | **27.50%** | **25.68%** | **24.06%** |
> |   CIFAR-100   |  WideResNet-34-10   | IAD-I  |   58.19%   |   40.34%   | **32.80%** | **31.03%** | **29.11%** |
> |   CIFAR-100   |  WideResNet-34-10   | IAD-II | **60.72%** | **40.67%** |   32.33%   |   30.40%   |   27.89%   |
> | Tiny-ImageNet | PreActive-ResNet-18 | IAD-I  |   46.66%   |   29.30%   |   22.68%   |   19.64%   |   17.82%   |
> | Tiny-ImageNet | PreActive-ResNet-18 | IAD-II | **49.52%** | **30.40%** | **23.42%** | **21.26%** | **18.64%** |
>
> **Table 2.** Comparison between IAD-I (IAD on basis of ARD) and IAD-II (IAD on basis of AKD$^2$) when the teacher network is trained by TRADES.
>
> |    Dataset    |       Network       | Method |  Natural   |    FGSM    |   PGD-20   |     CW     |     AA     |
> | :-----------: | :-----------------: | :----: | :--------: | :--------: | :--------: | :--------: | :--------: |
> |   CIFAR-10    |      ResNet-18      | IAD-I  |   80.62%   |   63.52%   | **53.83%** | **51.41%** | **50.15%** |
> |   CIFAR-10    |      ResNet-18      | IAD-II | **83.76%** | **64.17%** |   52.16%   |   50.59%   |   48.91%   |
> |   CIFAR-10    |  WideResNet-34-10   | IAD-I  |   82.89%   |   65.58%   | **55.86%** |   53.63%   | **52.29%** |
> |   CIFAR-10    |  WideResNet-34-10   | IAD-II | **85.68%** | **67.39%** |   55.45%   | **53.77%** |   51.85%   |
> |   CIFAR-100   |      ResNet-18      | IAD-I  |   54.75%   | **37.19%** | **30.71%** | **27.20%** | **26.10%** |
> |   CIFAR-100   |      ResNet-18      | IAD-II | **57.08%** |   36.94%   |   29.08%   |   25.83%   |   24.45%   |
> |   CIFAR-100   |  WideResNet-34-10   | IAD-I  |   56.97%   | **40.82%** | **33.60%** | **30.13%** | **29.05%** |
> |   CIFAR-100   |  WideResNet-34-10   | IAD-II | **60.82%** |   40.43%   |   32.39%   |   30.01%   |   28.01%   |
> | Tiny-ImageNet | PreActive-ResNet-18 | IAD-I  |   47.12%   |   29.12%   |   23.28%   |   19.52%   | **17.92%** |
> | Tiny-ImageNet | PreActive-ResNet-18 | IAD-II | **48.10%** | **30.54%** | **23.72%** | **19.78%** |   17.82%   |
>
> Regarding the variant of AKD$^2$ that replaces the second term in AKD$^2$ by our ''partially trust`` KL term (but without the introspection term), we find a similar improvement in the following table (Table 3).
>
> **Table 3.** Comparison between AKD$^2$ and a ''partially trust" variant with ResNet-18 on CIFAR-10.
>
> |                     |  Natural   |    FGSM    |   PGD-20   |     CW     |
> | :-----------------: | :--------: | :--------: | :--------: | :--------: |
> |       AKD$^2$       | **83.52%** |   63.91%   |   51.36%   |   50.36%   |
> | ''partially trust'' |   83.37%   | **63.95%** | **51.49%** | **50.40%** |

---

> ### Author Response · Authors · 2021-11-17
> **Response to Reviewer 2bpg: Part 2**
>
> > **Q2:** Most importantly, just like in ARD, IAD in Eq. (3) uses soft labels from teacher on clean images, instead of adversarial images. As I mentioned in bullet 1, the soft labels on clean images are generally trustworthy, as shown by yourself in the paper. So why does IAD partially trust it? It is never motivated, if I understand correctly.
>
> **A2:** In the following (Table 4), we conduct a range of experiments to compare IAD-I and its variant IAD-I-cons that replaces the downweighting with a constant 1.0 like ARD. According to the results, we can see that IAD-I-cons consistently achieves better robustness while sacrificing a little natural accuracy. We will use a constant 1.0 to replace the downweighting on $\ell_{kl}(S(\tilde{x}|\tau)||T(x|\tau))$.
>
> **Table 4.** Comparison between IAD-I-cons and IAD-I. IAD-I-cons is a variant of IAD-I, where the weight on $\ell_{kl}(S(\tilde{x}|\tau)||T(x|\tau))$ is a constant 1.0 like ARD.
>
> |    Dataset    |   Method   |  Natural   |    FGSM    |   PGD-20   |     CW     |
> | :-----------: | :--------: | :--------: | :--------: | :--------: | :--------: |
> |   CIFAR-10    | IAD-I-cons |   82.09%   |   63.20%   | **52.14%** | **50.74%** |
> |   CIFAR-10    |   IAD-I    | **83.33%** | **63.90%** |   51.77%   |   50.63%   |
> |   CIFAR-100   | IAD-I-cons |   55.53%   | **35.69%** | **27.40%** | **25.80%** |
> |   CIFAR-100   |   IAD-I    | **55.88%** |   35.68%   |   27.32%   |   25.60%   |
> | Tiny-ImageNet | IAD-I-cons |   46.14%   | **29.38%** | **23.40%** | **20.74%** |
> | Tiny-ImageNet |   IAD-I    | **46.66%** |   29.30%   |   22.68%   |   19.64%   |
>
> > **Q3:**  To support your intuitions, the current design of IAD in Eq. (3) might not be the best choice. Specifically, the "Student Introspection" KL loss is weighted by (1-$\alpha$). This is not so intuitive for me. You have justified the "Teacher Guidance" is not always trustworthy so it is good to weight its loss term with $\alpha$. However, a trustworthy "Teacher Guidance" doesn't mean an untrustworthy "Student Introspection". So why do you down-weight "Student Introspection" KL loss when $\alpha$ is large? In fact, the "Student Introspection" KL loss term is just an annealed version of the smoothness loss term in TRADES. As shown in TRADES paper, it regularizes the smoothness of the model which helps robustness. In my view, based on your key motivation to partially trust the teacher, the student introspection loss term should have a constant weight and instead of dynamically down-weighted as in Eq (3).
>
> **A3:** To verify this setting, we conduct an ablation study in both IAD-I and IAD-II and summarize the results in the following table (Table 5). According to the results, we can find there is no optimal coefficient for the student introspection term to achieve both the best natural accuracy and the best robustness. There is one trend that increasing the coefficient will gain more robustness with losing more natural accuracy. We will follow the reviewer's advice to set this coefficient as a hyperparameter $\gamma$, which could be flexibly instantiated by some constants or some strategic schedules to pursue the robustness or the natural accuracy.
>
> **Table 5.** Test accuracy of IAD-I and IAD-II using different coefficients for Student Introspection with ResNet-18 on CIFAR-10.
>
> |        |   coefficient   |   Natural   |   FGSM   |   PGD-20   |   CW   |
> | :----: | :-------------: | :---------: | :------: | :--------: | :----: |
> | IAD-I  |   1-$\alpha$    |   83.33%    |  63.90%  |   51.77%   | 50.63% |
> | IAD-I  |       0.5       |   81.26%    |  63.81%  |   52.61%   | 50.67% |
> | IAD-I  |       1.0       |   80.65%    |  63.71%  |   53.63%   | 50.87% |
> | IAD-I  |       2.0       |   78.20%    |  63.10%  |   55.21%   | 51.23% |
> | IAD-I  |       4.0       |   76.97%    |  63.05%  |   55.75%   | 51.07% |
> |        | **coefficient** | **Natural** | **FGSM** | **PGD-20** | **CW** |
> | IAD-II |   1-$\alpha$    |   83.21%    |  63.54%  |   51.85%   | 50.67% |
> | IAD-II |       0.5       |   82.76%    |  63.91%  |   52.50%   | 50.67% |
> | IAD-II |       1.0       |   81.33%    |  63.48%  |   53.32%   | 50.84% |
> | IAD-II |       2.0       |   80.95%    |  63.90%  |   54.02%   | 51.13% |
> | IAD-II |       4.0       |   80.43%    |  64.05%  |   55.27%   | 51.97% |

---

> ### Author Response · Authors · 2021-11-17
> **Response to Reviewer 2bpg: Part 3**
>
> > **Q4:** There are four loss terms in the best performing method AKD2+IDA. It would be nice if ablation studies can be provided on all those loss terms.
>
> **A4:** We term the combination of IAD and AKD$^2$ as IAD-II like Q1, and conduct the ablation study about four terms as follows. The experiments will be added to the Appendix.
>
> **Table 6.** The ablation study about the different terms in IAD-II with ResNet-18 on CIFAR-10. The $\checkmark$ means the term is considered in IAD.
>
> |            Label             |                      Self                      |                       Teacher$_{at}$                        |                       Teacher$_{st}$                        |             |            |
> | :--------------------------: | :--------------------------------------------: | :---------------------------------------------------------: | :---------------------------------------------------------: | :---------: | :--------: |
> | $\ell_{CE}(S(\tilde{x}), y)$ | $\ell_{KL}(S(\tilde{x}\|\tau)\|\| S(x\|\tau))$ | $\ell_{KL}(S(\tilde{x}\|\tau)\|\| T_{at}(\tilde{x}\|\tau))$ | $\ell_{KL}(S(\tilde{x}\|\tau)\|\| T_{st}(\tilde{x}\|\tau))$ | **Natural** | **PGD-20** |
> |         $\checkmark$         |                                                |                                                             |                        $\checkmark$                         | **84.30%**  |   50.67%   |
> |         $\checkmark$         |                                                |                        $\checkmark$                         |                                                             |   82.66%    |   51.72%   |
> |         $\checkmark$         |                                                |                        $\checkmark$                         |                        $\checkmark$                         |   83.52%    |   51.36%   |
> |         $\checkmark$         |                  $\checkmark$                  |                        $\checkmark$                         |                        $\checkmark$                         |   83.21%    | **51.85%** |
>
> Specifically, according to the table, we find $\ell_{KL}(S(\tilde{x}|\tau)|| T_{st}(\tilde{x}|\tau))$ can help model gain more natural accuracy, and $\ell_{KL}(S(\tilde{x}|\tau)|| T_{st}(\tilde{x}|\tau))$ can help model gain more robustness. AKD$^2$ achieves a good balance between two aspects by combining above two terms, and IAD-II further boosts the model robustness by incorporating the self-introspection term.
>
>
>
> In summary, we really appreciate the constructive advice of the reviewer. Regarding some experiments, we only improve a part of the model and run the experiments for ablation study. We will put all improvements on our method together to acquire the final results as soon as possible, and then upload the updated submission.

---

> ### Author Response · Authors · 2021-11-19
> **Need further clarification?**
>
> Thanks very much for your constructive comments on our work. We have tried our best to address the concerns. Is there any unclear point so that we should/could further clarify?

---

> > ### Comment · Reviewer_2bpg · 2021-11-22
> > **Thanks for your response**
> >
> > That solves my concerns. I've updated my score to 6. Thank you!

---

### Author Response · Authors · 2021-11-19
**General Response to All Reviewers**

We sincerely appreciate all reviewers' efforts in reviewing our paper. The draft is updated, where we revised our submission following the constructive comments from all reviewers and adding more ablation studies in Appendix.

We hope our responses below could address the reviewers' concerns.

Thank you.

The authors of Paper1426

---

### Decision · Program_Chairs · 2022-01-20

**Decision:**

Accept (Poster)

**Comment:**

This paper proposes a new knowledge distillation (KD) method for adversarial training. The key observation is inspiring: soft-labels provided by the teacher gradually becomes less and less reliable during the adversarial training of student model. Based on that,  they propose to partially trust the soft labels provided by the teacher in adversarial distillation.

Reviewers unanimously agree that this paper has clear motivation, well-sorted logic, and neat writing. While some reviewers initially posed concerns on evaluation completeness and detail clarification, they were well addressed during the rebuttal. AC reads the paper/discussion thread and agrees this is a worthy work to get accepted.